METHODS

# A molecular toolbox to modulate gene expression and protein secretion in the bacterial predator *Bdellovibrio bacteriovorus*

Ljiljana Mihajlovic[1], Lara M. Hofacker[1], Florian Lindner[1], Priyanikha Jayakumar[2], Andreas Diepold[3¤], Simona G. Huwiler[1]*

**1** Department of Plant and Microbial Biology, University of Zurich, Zurich, Switzerland, **2** Flow Cytometry Facility, University of Zurich, Zurich, Switzerland, **3** Department of Ecophysiology, Max Planck Institute for Terrestrial Microbiology, Marburg, Germany

¤ Current address: Institute of Applied Biosciences, Karlsruhe Institute of Technology, Karlsruhe, Germany
* simona.huwiler@uzh.ch

## Abstract

The predatory bacterium *Bdellovibrio bacteriovorus* kills and consumes other bacteria, thrives in diverse environments and holds great potential to address major challenges in medicine, agriculture, and biotechnology. As a bacterial predator it represents an alternative to traditional antimicrobial strategies to combat multidrug-resistant bacterial pathogens and prevent food waste, while the multitude of predatory enzymes it produces have potential for biotechnological applications. However, while a limited set of genetic tools exist, the lack of secretion assays and fine-tuning of secretion constrain both fundamental studies and bioengineering of *B. bacteriovorus*. Here, we present a molecular toolbox for *B. bacteriovorus* by systematically tuning gene expression and secretion of a reporter protein. Building on functional native and synthetic promoters from the Anderson library with varying expression levels of fluorescent reporter protein mScarletI3, we evaluated different ribosomal binding sites (RBS) to fine-tune gene expression. To examine secretion, we established a novel protocol to quantify extracellular release of a Nanoluc luciferase reporter protein in *B. bacteriovorus* using different native Sec-dependent signal sequences. We anticipate that the newly developed genetic toolkit and techniques will advance research on this fundamental predator-prey system, laying the foundation for its broader application and future bioengineering efforts. This work will pave the way for tailored applications of *B. bacteriovorus* in microbial ecology, agriculture, biotechnology, and medicine.

## Author summary

The predatory bacterium *Bdellovibrio bacteriovorus* kills and consumes other bacteria, including dangerous pathogens. It lives in a variety of environments

PLOS Genetics

**Data availability statement:** The Source data of this article can be downloaded from FigShare at https://doi.org/10.6084/m9.figshare.29565092.

**Funding:** This study was enabled by the Vontobel-Stiftung providing salary for L.M. (awarded to S.G.H.). During parts of the manuscript writing period the salary for L.M. was provided by Spark grant CRSK-3_228835 from Swiss National Science Foundation (SNSF) (awarded to L.M.). UZH Innovation grant Nr. BIOIG24-020 (awarded to L.M.) provided salary for L.M.H.'s summer internship and allowed critical flow cytometry experiments. F.L. was funded by a FreeNovation grant from Novartis Forschungsstiftung (awarded to S.G.H. and A.D.), and S.G.H. by SNSF Ambizione Fellowship PZ00P3_193401. A.D. was supported by the Max Planck Society. The funders had no rule in study design, data collection and analysis, decision to publish, or preparation of the manuscript.

**Competing interests:** The authors have declared that no competing interests exist.

and has great potential in medicine, agriculture, and biotechnology. Furthermore, the multitude of secreted predatory enzymes of *B. bacteriovorus* possess biotechnological potential. However, research and engineering efforts have been limited by a lack of secretion assay and fine-tuning of protein secretion tailored to this organism. Here, we present a molecular toolbox to modulate gene expression and protein secretion in *B. bacteriovorus*. We tested a range of native and synthetic gene promoters at population level and evaluated the effect of different ribosomal binding sites. Further, we established a protocol to quantify extracellular release of a reporter protein. By enabling more precise secretion control, our work brings *B. bacteriovorus* a step closer to practical use as a biological tool to address antibiotic resistance and other microbial challenges.

## Introduction

The microbial predator *Bdellovibrio bacteriovorus* has significant potential to address multiple challenges in medicine, agriculture and biotechnology [1–4]. It is a small predatory bacterium naturally present in numerous environments including soil [5], fresh water [6] and the human gut [7]. It has a unique predatory life cycle involving the killing, invasion and digestion of Gram-negative bacterial prey (S1 Fig). In the attack phase (AP), *B. bacteriovorus* approaches its prey by swimming in liquid or gliding on surfaces [8]. It then invades the periplasm of its prey, where it secretes hydrolytic enzymes that break down the prey components. These released prey nutrients fuel *B. bacteriovorus* growth and division, culminating in release of multiple progeny that continue the predatory cycle [9–11]. While significant progress has been made in understanding this intricate bipartite bacterial interaction, research is hindered by the limited tools available for modulating gene expression [12] and protein secretion in *B. bacteriovorus*.

The natural ability to target and eliminate multiple WHO priority bacterial pathogens and its inherent low immunogenicity [1,3] has positioned *B. bacteriovorus* as a promising alternative to traditional antimicrobial approaches, offering innovative solutions in multiple contexts [3,4,13]. *B. bacteriovorus* has potential as a 'living antibiotic' for combating multidrug-resistant infections [1,3]. Beyond clinical applications, it could also help to prevent or treat bacterial infections in farm animals [14], protect agricultural crops [15–18], and effectively disrupt or prevent bacterial biofilms [19,20]. Apart from its antimicrobial effects, *Bdellovibrio* species potentially play a role in maintaining balance in microbial communities [7,21] and serve as valuable models for studying microbial interactions and predator-prey dynamics [3,22,23].

Despite its enormous potential, *B. bacteriovorus* remains challenging to engineer due to the lack of versatile genetic tools compared with well-characterized bacteria like *Escherichia coli*. Without a comprehensive molecular toolbox for precise gene expression and protein secretion, efforts for controlled and targeted application of *B. bacteriovorus* remain limited. Engineering *B. bacteriovorus* is further complicated

by its dependence on prey for survival in the predatory life cycle and the limited availability of genetic tools. The range of independently replicating plasmids for *B. bacteriovorus* is narrow [24–27], and some functional, chemically inducible promoters (e.g., $P_{tac}$, $P_{JExD}$, $P_b$) as in *E. coli* have been described recently [12,27,28]. To date, three synthetic promoters from the Anderson collection have been characterized in *B. bacteriovorus* [27]. In addition, Kaljević et al. (2021) [29] employed the synthetic $P_{Biofab}$ promoter as a constitutive reference, further supporting the feasibility of synthetic promoter use in *B. bacteriovorus*. However, gene regulation mechanisms in *B. bacteriovorus* are largely unexplored [12]. The interaction of the promoter and ribosomal binding site (RBS) and their effects on tuning gene expression have not been explored. Exploring these different genetic elements via a molecular toolkit will increase our knowledge on gene expression and its regulation in *B. bacteriovorus*, enabling more precise control of gene expression in this unique predatory bacterium.

B. bacteriovorus* is versatile in secreting an extensive arsenal of hydrolytic enzymes encoded in its genome [30], with one of the largest fractions of secreted proteins in bacteria [4,31]. It is likely that the secretion of hydrolytic enzymes plays a crucial role in degradation and digestion once the predator is inside the prey's periplasm [30,32]. In addition, *B. bacteriovorus* secretes several proteases when encountering high-nutrient medium in attack-phase or on *S. aureus* biofilms [32,33]. Two of these proteases, Bd2269 and Bd2692, effectively removed *S. aureus* biofilms, when the purified proteins were added directly [34]. *B. bacteriovorus* encodes a type I and type II secretion system, as well as a twin-arginine translocation and complete Sec systems [30,35]. Although *B. bacteriovorus* holds great potential for the secretion of hydrolytic enzymes with biotechnological application [36], the secretion itself is largely unexplored, to the best of our knowledge. A contributing factor to this knowledge gap is the lack of established methods for rapid detection and quantification of protein secretion.

To address these challenges, we developed a molecular toolbox to modulate *B. bacteriovorus* gene expression and protein secretion. Specifically, we systematically evaluated native and synthetic promoters in combination with different ribosomal binding sites (RBSs), using a fluorescent reporter (mScarletI3) to quantify expression levels. We found that several synthetic promoters from the Anderson library [37] regulate *B. bacteriovorus* gene expression at different levels. Additionally, we demonstrate that promoter selection influences both intracellular expression of the fluorescent reporter mScarletI3 and extracellular secretion of the NanoLuc luciferase, with varying secretion efficiencies depending on the chosen native or synthetic promoters and secretion signals (S1 and S2 Tables). Collectively, these advances address existing limitations in *B. bacteriovorus* engineering and provide a robust framework for its application across diverse fields.

## Description of the method

### Bacterial cultivation

Wild-type *Bdellovibrio bacteriovorus* HD100[T], *Escherichia coli* S17-1, and *E. coli* S17-1 pZMR100 [38] were kindly provided by Prof. R. E. Sockett (University of Nottingham, UK). *E. coli* S17-1, and *E. coli* S17-1 pZMR100 were used for cultivation of *B. bacteriovorus* strains. *E. coli* NEB5α (New England Biolabs) was used for plasmid assembly via Gibson cloning.

Bdellovibrio bacteriovorus* HD100 strains were cultivated in 2 ml of Ca/HEPES buffer (5.94 g/l HEPES free acid, 0.294 g/l calcium chloride dihydrate, pH 7.6) with *E. coli* S17-1 or *E. coli* pZMR100 as prey at 29°C, shaking at 200 rpm. Kanamycin-resistant *B. bacteriovorus* strains carrying pCAT.000-derived plasmids [39] were grown with kanamycin-resistant *E. coli* pZMR100 in the presence of 50 µg/ml kanamycin. Wild-type *B. bacteriovorus* HD100 was cultured under the same conditions using *E. coli* S17-1 as prey, without kanamycin. For a lysate, 150 µl of stationary-phase prey (*E. coli* pZMR100 or *E. coli* S17-1, optical density measured at 600nm, $OD_{600}$ of 4.5) was added to Ca/HEPES buffer with *B. bacteriovorus*. Cultures were incubated for 24 hours, except for flow cytometry experiments, where incubation was extended to 48 hours. *B. bacteriovorus* revival plates were prepared according to Lambert & Sockett [40].

*E. coli* S17-1 strains were grown in YT broth at 37°C, shaking at 200 rpm. For kanamycin-resistant strains carrying pZMR100 [38] or strains with pCAT.000-derived plasmids [39], 25 µg/ml kanamycin was added. Phosphate-buffered saline (PBS, pH 7.4) was used for flow cytometry. A complete list of all *B. bacteriovorus* and *E. coli* strains used in this study is provided in S3 and S4 Tables (XLS table in online materials).

### Generation of *E. coli* and *B. bacteriovorus* strains with different reporter plasmids

**General cloning procedure.** Various reporter plasmids were constructed by fusing a pCAT.000-derived backbone [39] with different fluorescent proteins, promoter regions or protein secretion signals. The cloning strategies for these plasmids are detailed in the following sections. A complete list of plasmids and their compositions is provided in S5 Table.

**Generating reporter plasmids with different fluorescent proteins**. Coding regions for fluorescent proteins mCherry, mScarletI3 [41], mNeonGreen [42] were codon optimized for *B. bacteriovorus* (Twist Biosciences). Gene blocks with fluorescent protein coding regions were used as templates for amplification by PCR (98°C for 30 s; 30 cycles: 98°C for 10 s, 56.4°C for 10 s, 72°C for 45 s; 72°C for 5 min) using Phusion plus polymerase (Thermo Fisher) according to manufacturer's instructions, using primer pairs as listed in S6 Table.

Plasmid backbones, into which the coding regions for fluorescent proteins were inserted, were amplified from pFL015 (10 ng, S5 Table) using KOD Hot Start polymerase (Merck Millipore) (98°C for 10 s; 30 cycles: 98°C for 10 s, 57°C for 5 s, 68°C for 45 s; 72°C for 5 min) with primer pairs LM_pcat_fp_bb_fwd and LM_pcat_fp_bb_rev (S6 Table).

**Generating reporter plasmids to determine fluorescence intensity using different native and synthetic promoters and ribosomal binding sites.** All reporter plasmids contained a pCAT.000-derived vector carrying the codon-optimized mScarletI3 fluorescent protein pLH-C1 (S5 Table). Native promoter regions were amplified from B. *bacteriovorus* HD100 genomic DNA by PCR (98°C for 30 s; 30 cycles: 98°C for 10 s, 59°C for 15 s, 72°C for 30 s; 72°C for 5 min) using Phusion plus polymerase according to manufacturer's instructions using suitable primer pairs (S6 Table). Synthetic promoters from the Andersson library (http://parts.igem.org/Promoters/Catalog/Anderson) were ordered as oligonucleotides (Microsynth) and assembled by oligonucleotide annealing. To achieve this, 5 µl of each forward and reverse primer (S6 Table) was added into 90 µl HEPES buffer (3 mM, pH 7.5), mixed and heated to 95°C for 5 min, and cooled to 4°C (0.1°C/s). The annealed DNA fragments were diluted 1:10 and amplified by PCR under the same conditions as for native promoters (Primer details in S6 Table).

Plasmid backbones, into which the different promoter regions were inserted, were amplified from pLH-C1 (10 ng, S5 Table) using KOD Hot Start polymerase (94°C for 5 s; 30 cycles: 98°C for 10 s, 60°C for 5 s, 68°C for 90 s; 72°C for 5 min) and primer pairs as listed in S6 Table. PCR products were treated with DpnI (10 U) New England Biolabs (NEB) before Gibson assembly.

**Generation of plasmids for the Nanoluc secretion assay in *B. bacteriovorus*.** Reporter plasmids to monitor secretion in *B. bacteriovorus* were constructed using a pCAT.000-derived vector with the Nanoluc coding sequence as the template (pFL016, S5 Table). Secretion signal sequences were identified using SignalP 6.0 [43] (for details see S2 Table) and amplified from *B. bacteriovorus* HD100 genomic DNA by PCR (98°C for 30 s; 30 cycles: 98°C for 10 s, 60°C for 10 s, 72°C for 30 s; 72°C for 5 min) with Phusion polymerase (Primer details in S6 Table).

Plasmid backbones for insertion of secretion signal regions before the Nanoluc coding region were amplified from pFL016 (10 ng, S5 Table) using KOD Hot Start polymerase (98°C for 10 s; 30 cycles: 98°C for 10 s, 57°C for 5 s, 68°C for 45 s; 72°C for 5 min) and according to primer pairs (S6 Table).

**Gibson assembly and transformation of plasmids into *B. bacteriovorus*.** Different fluorescent protein sequences, promoter regions or secretion signal sequences were assembled into plasmid backbones originating from pCAT.000 [39] (for details see previous sections) using the Gibson assembly NEBuilder HiFi DNA Assembly Kit (New England Biolabs) following the manufacturer's instructions. 3 µl Gibson assembly reactions were transformed into chemically competent *E. coli* NEB5α cells which were plated on YT agar plates with 25 µg/ml kanamycin and incubated overnight at 37°C.

Plasmids were isolated from transformed *E. coli* NEB5α single clones and verified by Sanger sequencing (Microsynth, Balgach). Subsequently, the plasmids were transformed into *E. coli* S17-1 by heat shock for conjugation into wild-type *B. bacteriovorus* HD100. Conjugation was carried out as previously described [44].

Single plaques of the *B. bacteriovorus* strains containing the pCAT.000-derived plasmids were isolated and grown as described in the "bacterial cultivation" section on *E. coli* S17-1 pZMR100 with 50 µg/ml kanamycin. Glycerol stocks of *B. bacteriovorus* strains were generated by mixing 1.2 ml of a 24-h *B. bacteriovorus* lysate with 0.4 ml 80% sterile glycerol and immediate freezing in liquid nitrogen for long term storage at -80°C.

## Fluorescence quantification in *E. coli* and in *B. bacteriovorus* in attack phase (AP) using flow cytometry

mScarletI3 fluorescence in different strains of *E. coli* S17-1 and *B. bacteriovorus* HD100 AP were quantified using a 5-laser Cytek Aurora spectral flow cytometer. 1 ml of 48-hour cultures of stationary-phase *E. coli* S17-1 (grown at 200 rpm and 37° C) and *B. bacteriovorus* lysates (grown at 200 rpm and 29° C), were centrifuged (*E. coli*: $2'500 \times g$, 5 min; *B. bacteriovorus*: $5'000 \times g$, 20 min), resuspended in 1 ml PBS buffer, and diluted 1:100 in PBS.

mCherry, mScarletI3 or mNeonGreen fluorescence was measured separately in the Cytek Aurora detection channels YG3 (615/20 bandpass filter), YG2 (598/20 bandpass filter), or B2 (525/17 bandpass filter), respectively. Samples were acquired at a low flow rate (15 µl/min) and at an event rate of ~2000 events/sec. Threshold was set on the forward scatter (FSC) of 500 and side scatter (SSC) of 3000 and all fluorescence detector gains were set at 1500. Data acquisition was performed using SpectroFlo software (version 3.3.0; https://cytekbio.com/pages/spectro-flo) at the University of Zurich Flow Cytometry Facility. Doublets were excluded using SSC-A versus SSC-H gating. Bacterial cells were analysed using YG3-H, YG2-H or B2-H versus SSC-H density plots, and fluorescent population was defined using a gate set on the 95th percentile of the negative control (pCAT:P$_{merRNA}$-opt.RBS). Since there was no spectral overlap, no unmixing was performed, and the raw data was used for analysis.

Flow cytometry data were exported as .fcs files and analyzed in R (version 2024.09.0) using Bioconductor package, flowCore, for preprocessing and ggplot2 for visualization. Median fluorescence intensity within gated populations was calculated to evaluate protein expression.

## Fluorescence-based promoter activity quantification in *B. bacteriovorus* using plate reader assays

For temporally resolved measurements of transcriptional activity in different *B. bacteriovorus* strains, assay media was prepared by mixing 270 µl YT broth, 19.73 ml Ca/HEPES and 3.17 µl CaCl$_2$. In each well of a transparent sterile flat-bottom black 96-well plate (BRAND, Cat no.781671) we added 110 µl of assay media, 15 µl of *E. coli* S17-1 (pZMR100, with 50 µg/ml kanamycin) with an OD$_{600}$ of 4.50 (~$3.6 \times 10^9$ cells/ml) and 15 µl of 24-hour *B. bacteriovorus* lysate. 96-well plates were incubated at 29° C in a plate reader (Tecan Infinite M200) for 72 hours. OD$_{600}$ and mScarletI3 fluorescence (excitation: 540 nm, emission: 590 nm) were measured every 10 minutes. During incubation, plates were shaken continuously (double orbital, 2 mm) and covered with lids to minimize evaporation.

## Quantification of protein secretion in *B. bacteriovorus* AP using Nanoluc assay

Protein secretion in *B. bacteriovorus* AP was quantified using the Nano-Glo Luciferase Assay (Promega). 1 ml of 24-hour lysates were centrifuged in a 1.5-ml Eppendorf tube at $5'000 \times g$ for 20 min, and 50 µl of the supernatant was transferred to each well of a black 96-well plate (BRAND, Cat no.781671). The Nano-Glo Luciferase Assay Reagent was prepared by mixing Nano-Glo Luciferase Assay Substrate (1 volume) with Nano-Glo Luciferase Assay Buffer (50 volumes). Assay components were equilibrated to room temperature before use. 50 µl of the supernatant was combined with 25 µl of Nano-Glo Reagent in each well and mixed well. Luminescence was measured using plate reader (Tecan Infinite 200 Pro) with 200 ms integration time. Luminescence intensity, measured over a period of ~40 min, reflected total functional protein

secretion. To normalize Nanoluc secretion from viable *B. bacteriovorus* AP cells, the concentration of viable *B. bacteriovorus* AP cells was determined by viability stain and flow cytometry as described next.

## Viability assessment of *B. bacteriovorus* AP by flow cytometry

The viability of different *B. bacteriovorus* AP strains was assessed using the LIVE/DEAD BacLight Bacterial Viability and Counting Kit (Molecular Probes, L34856; Fisher Scientific) and a 5-laser Cytek Aurora spectral flow cytometer. 1 ml of the *B. bacteriovorus* lysates incubated for 24 hours at 29°C were centrifuged at 5'000 × *g* for 20 min, washed by resuspension in 1 ml PBS, and diluted 1:100 in PBS. The stock solutions of propidium iodide (PI, 20 mM) and Syto9 (3.34 mM) in the LIVE/DEAD BacLight Bacterial Viability and Counting Kit were diluted 1:10 in PBS buffer. From these dilutions 1 µl each was added to 200 µl of *B. bacteriovorus* AP containing sample and mixed well.

Fluorescence was detected using the YG3 channel (615/20 bandpass filter) and the B2 channel (525/17 bandpass filter) for PI- and Syto9-stained *B. bacteriovorus* AP cells. Samples were acquired at a low flow rate (15 µl/min) and at an event rate of ~1000 events/sec. Threshold was set on the FSC of 500 and SSC of 3000, and all fluorescence detector gains were set at 1000. Data were acquired using SpectroFlo software (version 3.3.0; https://cytekbio.com/pages/spectro-flo) at the University of Zurich Flow Cytometry Facility. The gating strategy excluded doublets (SSC-A vs. SSC-H). Further, PI- and Syto9-stained cells were identified using YG3-H vs. SSC-H and B2-H vs. SSC-H density plots. Live cells were distinguished from dead cells using YG3-H vs. B2-H density plots. A fixed sample volume (10 µl) was acquired, and the total number of viable cells was determined using the described gating strategy. The average number of viable cells per 1 µl was then calculated.

## Verification and comparison

### Establishment of a readout assay for promoter screening in *B. bacteriovorus* AP with mScarletI3 reporter

Our initial goal was to develop a modular expression vector to enable reliable assembly or exchange of genetic parts for tight control of gene expression in *B. bacteriovorus* AP. The vector was designed with a uniform architecture, incorporating a reporter gene preceded by genetic key regulatory components: a promoter and a ribosomal binding site (RBS) (Figs 1A and S2). To facilitate promoter screening, we generated a fluorescent readout system based on the pCAT.000 plasmid, a low-copy vector featuring an RSF1010-derived origin of replication [39]. This standardized design streamlines the screening process and provides a foundation for fine-tuning gene expression, facilitating the investigation of molecular mechanisms underlying predatory behavior and potential applications of *B. bacteriovorus* in various fields.

To identify an optimal reporter gene, we quantified the expression levels of the three different fluorescent reporter proteins mCherry, mNeonGreen [42], and mScarletI3 [41] in *B. bacteriovorus* AP cells - where they can be readily measured by flow cytometry- under the control of the strong native promoter $P_{merRNA}$ (Fig 1). The codon usage of all fluorescent reporter proteins was optimized for efficient expression in *B. bacteriovorus*, providing a reliable and quantitative readout of promoter and RBS activity. The empty plasmid control pCAT:$P_{merRNA}$-opt.RBS exhibited negligible fluorescence, confirming minimal background signal. Among the tested reporters, mScarletI3 displayed the highest fluorescence intensity, approximately 3-fold higher than mNeonGreen and 100-fold higher than mCherry (Fig 1B). These results highlight mScarletI3 as the optimal reporter for gene expression studies in *B. bacteriovorus* under these conditions, allowing for highly sensitive measurement and comparison of different promoter strengths.

### Different expression profiles driven by native *B. bacteriovorus* promoters

To expand the genetic engineering toolbox of *B. bacteriovorus*, we evaluated different types of promoters in *B. bacteriovorus* (Fig 2). For a systematic comparison, we tested native promoters for regulating gene expression initially (Fig 2A). Previous studies have shown that the native promoter $P_{merRNA}$ in *B. bacteriovorus* exhibits very high activity during the AP

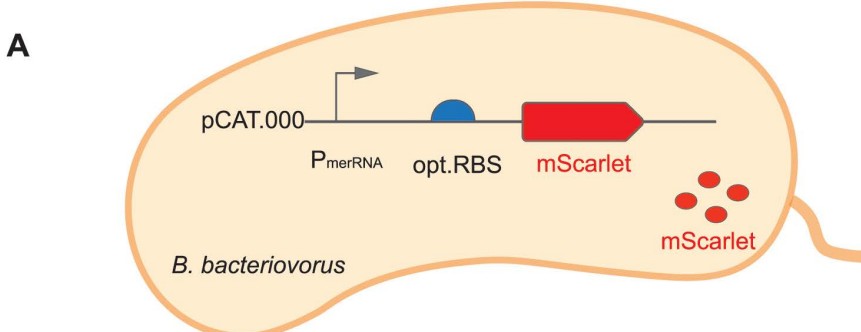

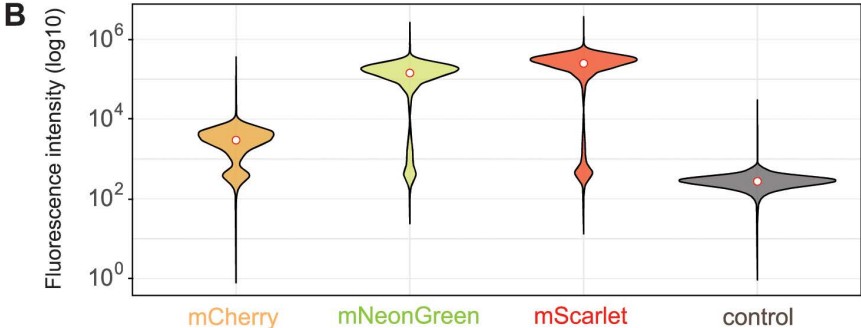

**Fig 1. Transcriptional readout assay comparing expression of fluorescent reporter proteins in *B. bacteriovorus* AP. (A)** Schematic representation of *B. bacteriovorus* AP cell expressing cytosolic mScarletI3 under the control of P$_{merRNA}$ promoter on a pCAT.000-derived plasmid. This figure was drawn by hand using Adobe Illustrator. **(B)** Comparison of fluorescence intensities of different fluorescent reporter proteins (mCherry, mNeonGreen, mScarletI3) expressed in *B. bacteriovorus* AP cells, measured at their respective emission wavelengths (610 nm, 517 nm, 592 nm, respectively). The control is plasmid pCAT:P$_{merRNA}$-opt.RBS lacking a fluorescent protein; its fluorescence was measured at 610 nm, 592 nm, and 517 nm, while a mean of all three measurements is shown here. White dots indicate median fluorescence. A second independent biological replicate produced highly similar results (see online source data).

[12,45]. Building on these findings, we leveraged RNA-seq data [45] to identify promoters associated with genes highly expressed during the attack phase (AP), reasoning that these promoters would likely drive strong transcription of heterologous genes. Approximately 67% of AP-specific genes in *B. bacteriovorus* are regulated by FliA (σ$^{28}$) [45], which is homologous to *E. coli* σ$^{28}$. In *E. coli*, FliA primarily governs flagellar and motility genes—accounting for roughly 1% of its genome [46] whereas in *B. bacteriovorus*, the same sigma factor controls a large fraction of AP genes, including many encoding surface proteins and proteases [45]. This expanded regulatory scope underscores the potential of FliA-driven promoters for high-level expression in *B. bacteriovorus* AP. We therefore selected several promoters with or without identifiable FliA binding motifs, including P$_{bd0149}$, P$_{bd1981}$, P$_{bd0064}$, P$_{bd1130}$, P$_{bd3548}$ (all with FliA motifs) and P$_{bd2209}$, P$_{bd3180}$ ("non-FliA") (Fig 2A). Each promoter region (~100 bp upstream of the putative translation start site, S1 Table) was placed upstream of a *B. bacteriovorus*-optimized ribosomal binding site (opt.RBS) [12,47] and the mScarletI3 reporter in the low-copy pCAT.000-derived plasmids.

Flow cytometry analyses of *B. bacteriovorus* AP cells harboring these plasmids revealed that most native promoters drove significantly higher fluorescence than the empty vector control (pCAT:P$_{merRNA}$-opt.RBS), although their strengths varied (Fig 2A). P$_{merRNA}$ was confirmed to be the strongest of the tested native promoters, in agreement with Dwidar *et al.* [12]. FliA-associated promoters P$_{bd0149}$, P$_{bd1981}$, P$_{bd0064}$ showed robust activity, while the other two P$_{bd1130}$, P$_{bd3548}$ showed moderate

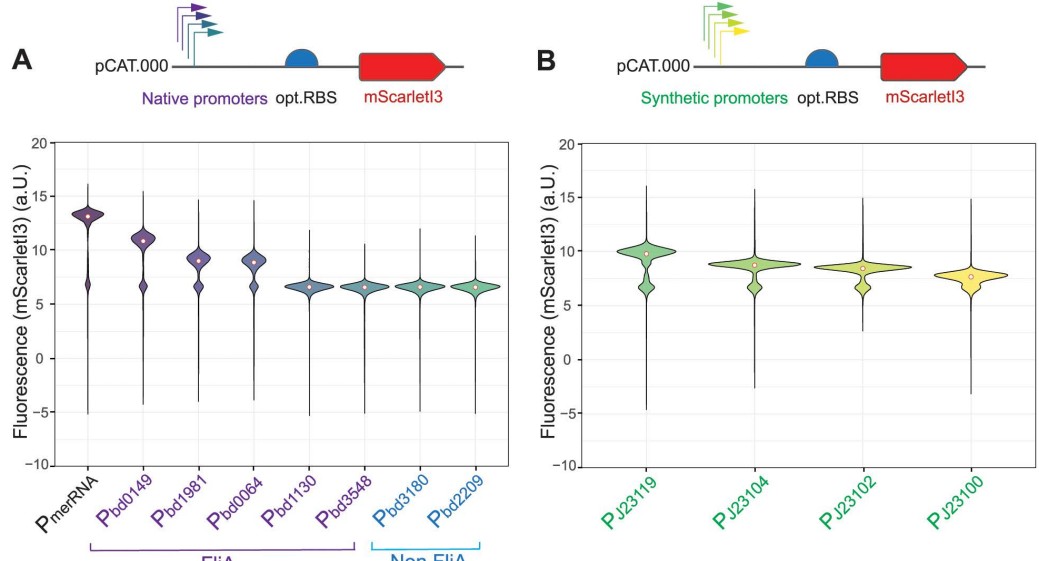

**Fig 2. Native and synthetic promoters exhibit distinct mScarletI3 expression levels and variability in *B. bacteriovorus* AP cells. (A)** Violin plots showing the distribution of mScarletI3 fluorescence for *B. bacteriovorus* AP cells carrying native promoters with FliA motifs ($P_{bd0149}$, $P_{bd1981}$, $P_{bd0064}$, $P_{bd1130}$, $P_{bd3548}$) or without FliA motifs ($P_{bd3181}$, $P_{bd2209}$). **(B)** Violin plots showing fluorescence levels driven by four synthetic Anderson promoters ($P_{J23119}$, $P_{J23104}$, $P_{J23102}$, $P_{J23100}$). Each promoter was tested in a pCAT.000-derived plasmid with an optimized RBS (opt.RBS) upstream of mScarletI3. Fluorescence intensity was measured using flow cytometry. White dots indicate median fluorescence. Positive mScarletI3 expression was defined by setting a gate at the 95th percentile of the fluorescence distribution of the negative control (pCAT:$P_{merRNA}$-opt.RBS, S3a Fig). The data show a single representative experiment out of two independent measurements with similar outcomes (see online source data).

expression levels. The two "non-FliA" promoters: $P_{bd3180}$ and $P_{bd2209}$ both produced only moderate fluorescence levels in *B. bacteriovorus* AP (Fig 2A).

Further, we tested the same set of native *B. bacteriovorus* promoters in the prey bacterium *E. coli* S17-1 to assess cross-species activity (S4a Fig). While $P_{merRNA}$ yielded robust fluorescence in *B. bacteriovorus* AP, it was notably weaker in *E. coli*, likely due to the specific function in the predator [45]. Overall, most *B. bacteriovorus* promoters drove moderate expression in *E. coli*, with $P_{bd0064}$ and $P_{bd3180}$ showing the highest expression levels. Interestingly, the two "non-FliA" promoters showed substantially greater expression in *E. coli* than in *B. bacteriovorus* AP (S4a Fig). This cross-species activity could be due to partial conservation of sigma factor motifs recognized by *E. coli* RNA polymerase. Their lower activity in *B. bacteriovorus* suggests that these ~100 bp regions upstream of the fluorescent reporter gene translational start site may lack *B. bacteriovorus* specific regulatory elements.

Overall, our analysis confirms that multiple native promoters can drive strong gene expression in *B. bacteriovorus* during the AP, while promoters with FliA motifs seem to have a higher promoter activity. However, because these native promoters are not fully orthogonal to *B. bacteriovorus*' endogenous regulatory network, using them additionally on a plasmid may cause unintended cross-talk with the same promoters in the genome. Consequently, we next investigated four synthetic promoters to achieve more independent, fine-tuned control of gene expression.

### Four synthetic promoters from the Anderson library are functional in *B. bacteriovorus* AP

To broaden the range of expression levels available in *B. bacteriovorus*, we evaluated four synthetic promoters derived from the widely used Anderson library (http://parts.igem.org/Promoters/Catalog/Anderson) (Fig 2B). These promoters (e.g., $P_{J23119}$, $P_{J23104}$, $P_{J23102}$, $P_{J23100}$) are based on a $\sigma^{70}$ consensus sequence and have been successfully employed in

*E. coli* for tunable gene expression [48]. Promoter $P_{J23119}$ was included, since it was used as a benchmark in other promoter characterization studies [49,50]. We observed varying expression levels among these synthetic promoters in *E. coli* S17-1, with $P_{J23119}$ exhibiting the highest activity (S4b Fig). When tested in *B. bacteriovorus* AP the Anderson promoters consistently showed moderate-to-high expression with different relative strengths (Fig 2B). $P_{J23119}$ generated the highest, $P_{J23104}$ and $P_{J23102}$ an intermediate, whereas $P_{J23100}$ produced the lowest fluorescence level of mScarletI3. Our results align with Salgado et al. [27] in showing graded activity of Anderson promoters in *B. bacteriovorus*, with the shared $P_{J102/}$ $P_{J23102}$ promoter performing at an intermediate level across studies. Differences in precise rank ordering beyond that overlap are expected given differences in reporter/context (mScarletI3 vs mRFP1), genetic architecture and measurement setup. By expanding the panel (adding $P_{J23119}$ at the strong end and $P_{J23100}$ at the weak end) our work extends prior characterizations to a broader dynamic range and application space. Notably, the strong activity of $P_{J23119}$ in *B. bacteriovorus* AP suggests the conservation of the transcriptional machinery between *E. coli* and *B. bacteriovorus*, at least for some $\sigma^{70}$-like regulated promoters. *B. bacteriovorus* RNA polymerase $\sigma^{70}$ (Bd0242) [30] could effectively recognize $\sigma^{70}$ (-like) promoters, consistent with the known conservation of core $\sigma^{70}$ consensus elements across many proteobacteria [51]. While the exact mechanism remains to be elucidated, particularly regarding the role of any *B. bacteriovorus*-specific $\sigma$ factors, our data indicate that carefully chosen synthetic promoters can be used for reliable and tunable gene expression.

### Differential expression patterns of native and synthetic promoters in *B. bacteriovorus* during predation

To investigate how native versus synthetic promoters influence *B. bacteriovorus* gene expression over time, we monitored mScarletI3 fluorescence in predator–prey mixtures using a plate reader for up to 72 hours. *B. bacteriovorus* carried pCAT.000-derived plasmids encoding mScarletI3 under either native promoters ($P_{bd0064}$, $P_{bd0149}$, $P_{bd1981}$) or synthetic Anderson promoters ($P_{J23119}$, $P_{J23102}$, $P_{J23104}$) with *B. bacteriovorus* optimized RBS [12]. Dynamics at early time points (0–10 hours) (Fig 3A) showed that cultures expressing mScarletI3 under the synthetic promoters ($P_{J23119}$, $P_{J23102}$, $P_{J23104}$) tend to display a faster increase in mean fluorescence. By comparison, native promoters ($P_{bd0064}$, $P_{bd0149}$, $P_{bd1981}$) remained at unchanged levels until approximately 4 hours after mixing predator and prey. This delay is consistent with native regulatory elements active during the AP of *B. bacteriovorus* after being released from the prey. In parallel, the $OD_{600}$ measured during the predation of different *B. bacteriovorus* strains on *E. coli* declined steadily at similar rates (Fig 3B), likely reflecting a very similar lysis speed of the *E. coli* prey population.

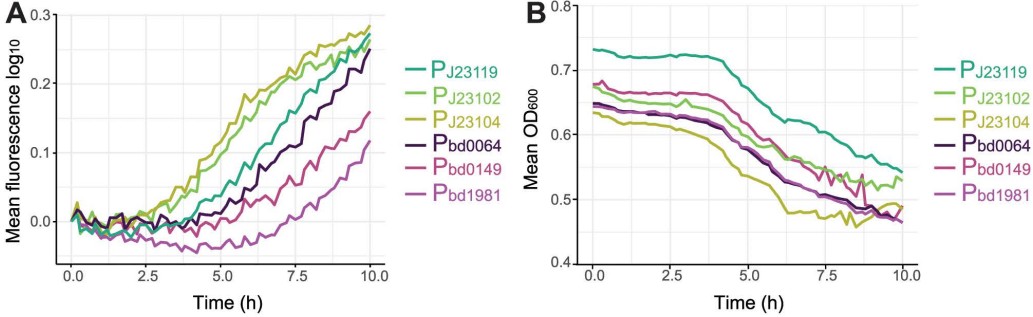

**Fig 3. Temporal gene expression of selected synthetic and native promoters in *B. bacteriovorus* populations during predation. (A)** Mean mScarletI3 fluorescence (590 nm) measured over the first 10 hours of mixing *E. coli* S17-1 and *B. bacteriovorus* with native ($P_{bd0064}$, $P_{bd0149}$, $P_{bd1981}$) or synthetic ($P_{J23102}$, $P_{J23104}$, $P_{J23119}$) promoters expressed from pCAT.000-derived plasmids. Each promoter was tested in a pCAT.000-derived plasmid with an optimized RBS (opt.RBS) upstream of mScarletI3. For clarity of comparison, fluorescence starting values were adjusted to a common baseline across all samples. **(B)** Corresponding $OD_{600}$ profiles over time, illustrating prey cell lysis of *E. coli* S17-1 by the same *B. bacteriovorus* strains as in **a**. Data represents the average of two biological replicates, each measured in two technical replicates.

To examine the expression patterns over a longer time in *B. bacteriovorus* AP we continued monitoring the fluorescence of the cultures for up to 72 hours (S5 Fig). The native promoters display a more gradual increase in fluorescence during the early stages of the predatory life cycle, consistent with a regulatory pattern tightly linked to *B. bacteriovorus*-specific gene expression dynamics [45].

While synthetic promoter $P_{J23119}$ reached similar fluorescence end levels as the native promoters, fluorescence levels in the strains with $P_{J23102}$ and $P_{J23104}$ seem to slowly decrease from about 24-hr on. While this suggests that the heterologous $P_{J23119}$ promoter remains active for an extended period in newly formed *B. bacteriovorus* AP cells, the fluorescence observed with $P_{J23102}$ and $P_{J23104}$ at later time points may result from mScarletI3 protein that was produced during the initial ~12 hours of growth on *E. coli* S17-1. Despite these differences in timing, all tested promoters maintained discernible fluorescence over the 72 hours. In all predator-prey cultures, $OD_{600}$ values declined steadily after an initial rapid decrease and eventually stabilized at low levels, likely reflecting the lysis dynamics of the *E. coli* prey population. Overall, we show that synthetic promoters drive robust, early-onset expression in *B. bacteriovorus*, largely independent of its life cycle phase, unlike native AP promoters, which appear more tightly regulated. To achieve a more fine-tuned expression regulation we next investigated the effects of different RBS sequences.

## Optimization of the RBS generally enhances gene expression of native promoters in *B. bacteriovorus* AP

RBSs play a pivotal role in translation initiation by guiding the ribosome to the correct start codon on the mRNA, thereby influencing overall protein expression levels. Even slight alterations in RBS sequences, particularly in the Shine-Dalgarno sequence (SD) [52] and surrounding regions, can markedly affect translational efficiency. Recent work in various bacteria has highlighted the power of RBS engineering to tune gene expression for synthetic biology and metabolic engineering applications [47]. In *B. bacteriovorus*, however, RBS optimization is still an emerging area of research [12].

To test the importance of the RBS in *B. bacteriovorus*, we compared plasmids harboring either an optimized RBS (opt. RBS) [12] or no RBS for several promoters ($P_{merRNA}$, $P_{bd0064}$, $P_{bd0149}$, and $P_{J23119}$). As expected, fluorescence distribution of *B. bacteriovorus* AP populations revealed that the absence of RBS drastically lowered mScarletI3 fluorescence for both native and synthetic promoters (S6 and S7 Figs). As expected the expression level without RBS is moderate to low in *B. bacteriovorus* AP (S6 Fig), while the expression of synthetic promoter $P_{J23119}$ seems to be less affected by the absence of RBS (S6 Fig). When we compare the absence of RBS in synthetic promoter $P_{J23119}$ in *E. coli* the proportion of cells expressing the gene is higher than in *B. bacteriovorus* AP, with overall a low level fluorescence expression (S3 and S7 Figs). This finding agrees with prior observations in *E. coli* [53].

Building on this, we evaluated how an optimized RBS derived from a previous *B. bacteriovorus* study [12] compares to the native RBS sequences embedded within ~100 bp upstream of the start codon in four native promoters ($P_{bd0064}$, $P_{bd0149}$, $P_{bd1981}$, and $P_{bd3180}$). When placed upstream of the sequence encoding mScarletI3, the optimized RBS boosted protein expression compared to the native RBS in three out of four cases evaluated (Fig 4A). Interestingly, while the native sequences upstream of *bd0064, bd0149* and *bd1981* drive lower expression, incorporating optimized RBS elements significantly enhances their expression levels (Fig 4A). In contrast, the native sequence upstream of *bd3180* drove higher expression levels than its counterpart with an optimized RBS. This discrepancy may result from altered spacing in the native $P_{bd3180}$ promoter region, while the optimized RBS could have introduced suboptimal local mRNA structures that reduced translational efficiency.

To assess whether RBS sequences designed for *E. coli* also function well in *B. bacteriovorus* AP, we tested synthetic RBS (syn.RBS) BBa-B0034, a widely used strong RBS optimized for *E. coli* (iGEM Registry, http://parts.igem.org/Part:BBa_B0034), alongside the *B. bacteriovorus*-optimized RBS (opt.RBS) [12] for both a synthetic Anderson promoter ($P_{J23119}$) and a native promoter ($P_{bd0149}$). As shown in Fig 4B, combining the synthetic promoter $P_{J23119}$ with a synthetic RBS resulted in a slight increase in mScarlet fluorescence compared to the combination with a *B. bacteriovorus*-optimized

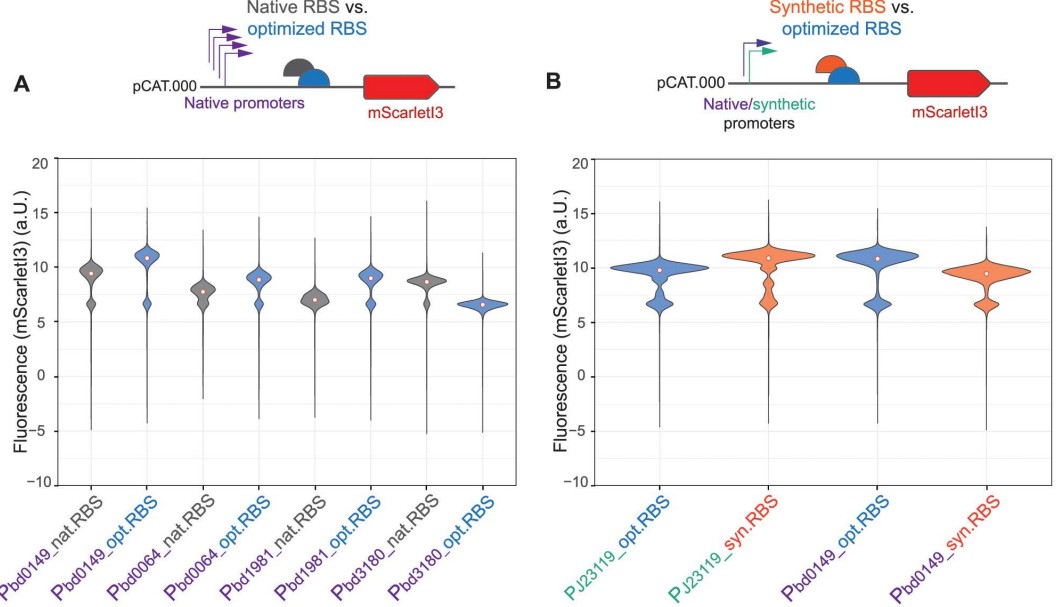

**Fig 4. Influence of Ribosomal Binding Site (RBS) modifications on *B. bacteriovorus* AP gene expression. (A)** Comparison of native versus optimized RBS sequences for four native promoters ($P_{bd0149}$, $P_{bd0064}$, $P_{bd1981}$, $P_{bd3180}$) **(B)** Comparison of *B. bacteriovorus*-optimized RBS (opt.RBS) [12] (as in Fig 2) versus *E. coli*-optimized synthetic RBS (syn.RBS) for both the synthetic promoter $P_{J23119}$ and the native *B. bacteriovorus* promoter $P_{bd0149}$. Fluorescence was measured in *B. bacteriovorus* AP populations using flow cytometry. White dots indicate median fluorescence (density plots shown in S3a Fig). A second biological and independent repeat of these measurements showed the same outcome (see source data).

RBS. This outcome is expected, as both the $P_{J23119}$ promoter and the synthetic RBS were originally optimized for *E. coli*. In contrast, the native *B. bacteriovorus* promoter $P_{bd0149}$ with the synthetic RBS led to a decrease in mScarlet fluorescence compared to the same promoter that was combined with the species-optimized RBS. These results highlight the importance of species-specific RBS optimization for achieving efficient gene expression in *B. bacteriovorus* AP.

In summary, our data confirms the critical influence of the RBS on protein expression in *B. bacteriovorus* AP. Removing the RBS seems to drastically lower detectable protein output, and replacing the native RBS sequences with *B. bacteriovorus*-optimized versions significantly boosts expression for most native promoters tested. Furthermore, an RBS optimized for *E. coli* is less efficient than the *B. bacteriovorus*-optimized RBS version when combined with a native promoter highlighting the need for species-specific optimization of the RBSs in this predatory bacterium.

## Application

### Fine-tuning Sec-dependent protein secretion in *B. bacteriovorus* AP

Protein secretion in *B. bacteriovorus* holds promise for targeted degradation of prey bacteria and/or their biofilms [33,34] and potential biotechnological applications [4], yet a sensitive quantitative assays and fine-tuned secretion of a target protein have been lacking. To address this gap, we developed a luciferase-based secretion assay for AP cells, using different promoters to drive expression of the NanoLuc luciferase [54] (Fig 5) and applied the different promoters with an optimized RBS, which were tested previously for intracellular protein expression. As a reporter protein for secretion, NanoLuc fused to an N-terminal signal peptide (ss_Bd2692$_{1-20}$, first 20 amino acids of Bd2692) was secreted into the supernatant to catalyze a luminescent reaction with furimazine and oxygen, enabling measurement of secretion directly from supernatants (Fig 5A).

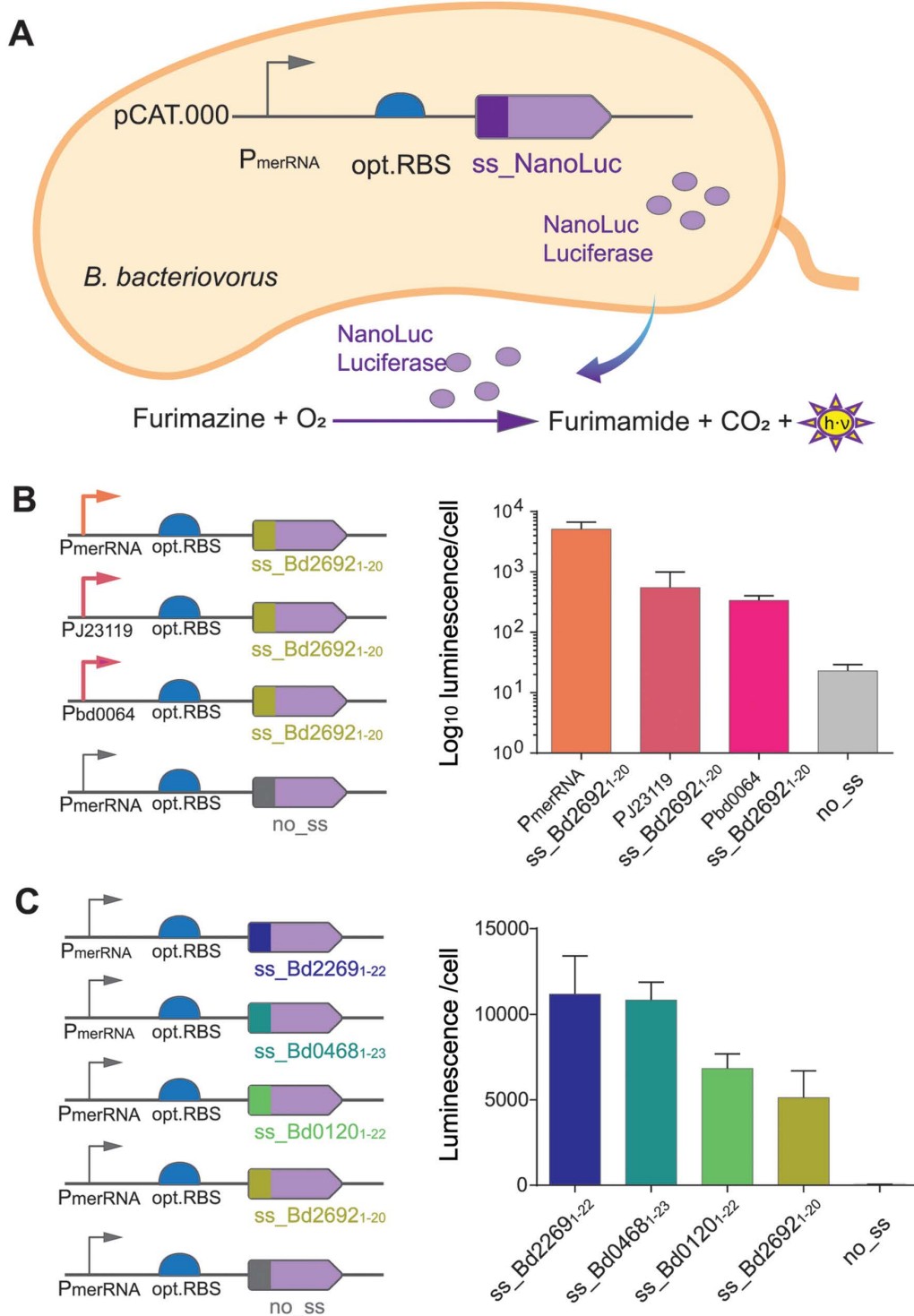

**Fig 5. NanoLuc secretion assay in *B. bacteriovorus* AP** (A) Schematic representation of the plasmid construct in *B. bacteriovorus* AP cells, showing NanoLuc luciferase fused to an N-terminal sec-dependent signal sequence (ss_NanoLuc) under the control of $P_{merRNA}$. The secreted NanoLuc reacts with furimazine, producing detectable luminescence (h·v). The no_ss control without secretion signal served as background signal for internal, non-secreted NanoLuc. Bars represent mean luminescence with indicated standard deviation detected in culture supernatants (two technical replicates); a second independent biological replicate yielded similar results (see source data). (B) Effect of promoters with different strengths ($P_{merRNA}$, $P_{J23119}$, or $P_{bd0064}$) on NanoLuc secretion using the ss_Bd2692$_{1-20}$ signal peptide. (C) Comparison of four signal peptides ss_Bd2269$_{1-22}$, ss_Bd0468$_{1-23}$, ss_Bd0120$_{1-22}$, and ss_Bd2692$_{1-20}$, in driving extracellular NanoLuc secretion. Fig 5A was drawn by hand using Adobe Illustrator.

## Promoter strength modulates the level of Sec-dependent reporter protein secretion

Based on the tested panel of promoters, we placed strong native promoter $P_{merRNA}$, the synthetic $P_{J23119}$ or the moderate native $P_{bd0064}$ upstream of signal peptide ss_Bd2692$_{1-20}$ (Fig 5B, left panel). Luminescence measurements from supernatants revealed that promoter strength correlated directly with the level of secreted NanoLuc (Fig 5B, right panel). $P_{merRNA}$ showed highest extracellular NanoLuc (e.g., $5.1 \times 10^3$ a.U.), whereas $P_{J23119}$ and $P_{bd0064}$ were ~13 fold lower ($4 \times 10^2$ a.U. and $3 \times 10^2$ a.U.). As expected, removing the signal peptide (no_ss) effectively abolished secretion (minimal background 23 a.U.), indicating that the Sec pathway can be used for export of NanoLuc in *B. bacteriovorus* AP with an according native Sec-dependent signal peptide. By fine-tuning the promoter choice, we modulated secretion levels from high to moderate, creating a flexible system for engineering *B. bacteriovorus* as a protein-delivery chassis.

## Different Sec-dependent signal peptides confer variable secretion efficiency

To further refine control over secreted protein levels, four different Sec-dependent signal peptides, ss_Bd2269$_{1-22}$, ss_Bd0468$_{1-23}$, ss_Bd0120$_{1-22}$ and ss_Bd2692$_{1-20}$ (S2 Table) were tested to quantify secretion from *B. bacteriovorus* AP under identical conditions using the strong native promoter $P_{merRNA}$ for transcription (Fig 5C, left panel).The signal peptide of Bd0468 was used as a positive control since it was shown previously that Bd0468-mCherry is secreted by *B. bacteriovorus* into the Bdelloplast by epifluorescence microscopy [55]. Depending on the signal peptide, different levels of extracellular NanoLuc were detected (Fig 5C, right panel). ss_Bd2269 and ss_Bd0468 exhibited the highest luminescence ~$1.1 \times 10^4$ a.U. ss_Bd0120 showed an intermediate mean value of $6.8 \times 10^3$ a.U., while ss_Bd2692 yielded a moderate yet significant secretion ($5.1 \times 10^3$ a.U.) above background. In contrast, the negative control lacking any signal peptide (no_ss) displayed only minimal luminescence confirming that successful export depends on a functional Sec signal. These data confirm that *B. bacteriovorus* AP can secrete heterologous proteins through the Sec-dependent secretion pathway and that choosing an appropriate signal peptide is critical for maximizing secretion efficiency.

Taken together, we established a versatile NanoLuc-based secretion assay by demonstrating that different native and synthetic promoters, next to different endogenous Sec-dependent signal peptides can direct the amount of heterologous proteins secreted by *B. bacteriovorus* AP. Beyond serving as a convenient readout for secretion studies, this platform paves the way for using *B. bacteriovorus* as a delivery chassis for secretion of enzymes or proteinaceous antimicrobial effectors in situ. This opens new avenues for targeted biocontrol and other biotechnological applications.

## Discussion

In this study, we have developed a molecular toolbox for the predatory bacterium *B. bacteriovorus* which enables precise control of both gene expression and protein secretion. By systematically evaluating native and synthetic promoters, multiple ribosomal binding sites (RBSs), and Sec-dependent signal peptides, we show that *B. bacteriovorus* can be engineered with far greater versatility than previously possible reducing the gap to more established model organisms such as *E. coli*. Crucially, we have integrated single-cell analyses (using flow cytometry) and population-level assays (using a plate-reader for time-course fluorescence assays and a NanoLuc-based secretion assay) to capture the inherent complexity and heterogeneity of the organism. This integrated approach highlights the potential of *B. bacteriovorus* for sophisticated engineering and lays the groundwork for advances in both fundamental research and biotechnological applications.

A key finding of our study is the identification of several promoters, both native and synthetic, that provide robust and tunable gene expression in *B. bacteriovorus* during the AP. Promoters containing a putative FliA ($\sigma^{28}$) binding motif tended to yield stronger AP-specific expression, consistent with the central role of $\sigma^{28}$ in motility and other AP-related functions [45] (Fig 2A). In contrast, non-FliA promoters that were highly active in *E. coli* stationary-phase were expressed only at a very low level in *B. bacteriovorus* AP (Figs 2A and S4), highlighting the distinct regulatory architecture of *B. bacteriovorus* and reinforcing the need to validate promoter functionality in the target organism.

Our results are in line with those of Salgado et al. [27], and in addition we expand the dynamic range of Anderson promoters characterized in *B. bacteriovorus* ($P_{J23119}$, $P_{J23104}$, and $P_{J23100}$). Notably, synthetic σ70-based promoters from the Anderson library [56,57] sometimes outperformed native promoters in both timing (earlier onset in the predation cycle) and level of expression in *B. bacteriovorus* (Figs 2B and 3A). This suggests a partial conservation of the transcriptional machinery between *B. bacteriovorus* and more conventional Gram-negative bacteria, despite the predatory life cycle of *B. bacteriovorus* with specialized developmental stages. Similar findings in other bacteria like *Myxobacteria*, *Cyanobacteria* and *Actinobacteria*, indicate that well-chosen synthetic promoters can function across phylogenetically distant bacteria, at least for σ70-dependent transcription [56–58]. While our study primarily examined how native and synthetic promoters regulate downstream gene expression in *B. bacteriovorus* AP, the question of interchangeability of specific promoters between different bacterial species, and in this case even the predator-prey gap, requires further investigation. In particular, FliA—which acts as a master regulator in *B. bacteriovorus* has a narrower regulatory scope in *E. coli*, underscoring how sigma factor homology may not guarantee identical promoter recognition across species.

Our results show that *B. bacteriovorus* AP can recognize multiple promoter types, in agreement with Salgado et al. [27], likely through the activity of its conserved σ70 apparatus and a specialized σ28 factor (FliA, gene *bd3318* [45]). *B. bacterivorus* encodes one σ70 homologue (gene *bd3314*) and one σ70 factor RpoD (gene *bd0242*) [59]. Notably, RpoD shares high similarity in its DNA-binding subdomains with the RpoD of *E. coli* K-12 [27], which may explain why *B. bacteriovorus* RpoD can recognize and drive expression from synthetic σ70-dependent *E. coli* promoters [27]. The ability to integrate different regulatory cues sets the stage for detailed studies of how global regulators (such as c-di-GMP, cGAMP, and potential other signaling molecules [59–63]) control the transition between different phases of the predatory life cycle. By mapping these networks, we can better understand the complex predatory life cycle of *B. bacteriovorus* and develop more precise genetic tools for controlling gene expression. We hope that in the future condition-responsive promoters may allow fine-tuned timing of predatory functions, which would help to decipher the intricate predator–prey interactions across multiple phases.

To better understand how native and synthetic promoters function within individual *B. bacteriovorus* cells, we needed to address the challenge of capturing expression variability at the single cell level. Such variability is often overlooked in population-level analyses, especially for relatively small *B. bacteriovorus* cells. In this study, we used flow cytometry to measure fluorescence in individual cells, to detect how different promoters and RBSs effect the expression level in *B. bacteriovorus* and compare it to *E. coli* and elucidate the proportion of cells in ON or OFF expression state (S7 Fig).

Salgado et al. [27] showed that the Anderson promoters $P_{J102}$ and $P_{J116}$ followed by synthetic bicistronic RBS BCD2 [64], shows a prominent heterogeneity of expression in a pSEVA plasmid with gentamycin resistance expressing mRFP1. The latter was detected by microscopy and confirmed by flow cytometry measurements in two clearly distinct populations regarding two distinct fluorescent protein expression levels. In contrast, the pCAT-based plasmids in this study showed no such prominent heterogeneity, but rather a natural distribution of different expression levels. This is a good pre-requisite to express and secrete high amounts of specific proteins in a defined way. The only exception to this are the minor subpopulations for the pCAT-plasmids expressed in *B. bacteriovorus* AP with $P_{J23119}$ and the synthetic RBS or optimized RBS (Figs 4 and S3). As the pCAT-plasmids have an RSF1010-derived ori like the pSEVA plasmids, a potential difference of expression might have been caused by other differences in the plasmids like for example the local plasmid context and the type of selective antibiotics as in this study kanamycin was used.

Although Salgado et al. [27] used a different cloning technique (hierarchical assembly cloning technique Golden Standard based on the Golden Gate modular cloning) than our approach (Gibson Assembly), the overall result is the same on achieving different levels of gene expression based on different promoters. While the expression of target proteins is stronger on a multi-copy plasmid and faster in the assembly process, if a target protein should be stably maintained in *B. bacteriovorus* without antibiotic selection pressure a chromosomal insertion is preferable [27]. Therefore our system is most useful in the context of basic research, where the presence of additional antibiotics is not as relevant or where

higher overall amounts of target proteins based on the multiple copies per cell are important. Although the copy number of the IncQ family RSF1010 replicon plasmids has not been measured in *B. bacteriovorus*, it is maintained at around ten copies per chromosome in at least two different Gram-negative bacteria [26,65]. It is known that chromosomal integration of a target gene region in *B. bacteriovorus* can be achieved by double homologous recombination [66,67] or Tn7-mediated chromosomal integration [27]. While the Tn7-mediated integration is faster to achieve, it is currently restricted to integrate downstream of *glmS*, a glutamine-fructose-6-phosphate aminotransferase [27] within Bd3423 with unknown function.

Building on our promoter-based transcriptional control, we revealed that small changes in the 5′ untranslated region can have a profound effect on protein production levels. An RBS optimized for *E. coli* was less efficient to an RBS tailored for *B. bacteriovorus* (Fig 4B), underscoring the importance of species-specific design. For most native promoters, replacing the short endogenous RBS with this optimized sequence resulted in a substantial increase in protein expression (Fig 4A). These data highlight that translational control in *B. bacteriovorus* can be fine-tuned by engineering the Shine-Dalgarno sequence and surrounding region, a principle that has been exploited in other bacterial systems [68]. Combined with careful promoter selection, RBS engineering offers a powerful dual strategy for achieving precise and high-level gene expression in this predatory bacterium.

By pairing promoter testing with secretion readouts, our work extends prior characterizations [12,27] to a broader dynamic range and application space. A major outcome of this work is a secretion assay for *B. bacteriovorus* that provides a rapid and sensitive method for measuring extracellular protein levels. By pairing NanoLuc luciferase with several Sec-dependent signal peptides derived from *B. bacteriovorus* proteins, we identified clear differences in secretion efficiencies. Some signal peptides produced high extracellular NanoLuc activity, whereas complete removal of the signal peptide abolished secretion (Fig 5C). By comparing different native Sec-dependent signal peptides, we establish an initial framework for assessing how signal sequence choice influences secretion efficiency, which can be expanded in future studies. In addition, different promoters affected how much NanoLuc was ultimately exported (Fig 5B), highlighting that both transcriptional strength and post-translational signals shape protein secretion. While in this assay NanoLuc luciferase was used as heterologously expressed and secreted protein, we aim to expand the repertoire of heterologous secreted proteins. The later will allow us to gain further insights into the secretion pathway in *B. bacteriovorus* while understanding its limitations.

This assay not only provides a simple and straightforward readout to study *B. bacteriovorus* secretion, but also opens the door to engineering the bacterium as a protein delivery vehicle. Its ability to secrete hydrolytic enzymes or other bioactive proteins could be exploited for biotechnological and therapeutic applications, such as targeted biofilm disruption, crop protection, or antimicrobial strategies against multidrug-resistant pathogens [1,4,15,18,19,36]. In addition, NanoLuc-based secretion assays could help to understand alternative secretion routes, such as the Twin-arginine translocation pathway. Further, our assays set the groundwork to extend this assay by using "split" luciferase reporters, which would enable spatio-temporal insights into secretion events within the prey cells. These advances may further enhance our ability to rationally design *B. bacteriovorus* as a versatile delivery chassis in different settings.

From an applied perspective, the ability to tune protein production and secretion in *B. bacteriovorus* holds great promise for exploiting its natural predatory traits against a range of pathogens. Engineered strains could deliver additional antimicrobial effectors directly to prey cells, degrade persistent biofilms more efficiently, or serve as on-site biocontrol agents in agriculture. In addition, *B. bacteriovorus*' outer membrane containing an atypical LPS [69] and its inherent low immunogenicity [1] support its potential as a 'living antibiotic'. In the future, engineered predatory bacteria could also work alongside existing antimicrobial treatments, e.g., by increasing their effectiveness, reducing the dose required, and/or prevent emerging resistance. Understanding how *B. bacteriovorus* behaves under varied environmental or host conditions, such as limited nutrients or distinct microbiome compositions, will be critical to advance its therapeutic and biocontrol applications. The genetic tools and assays developed here, which enable tunable gene expression and robust secretion measurements, provide a foundation for probing how *B. bacteriovorus* adapts to these diverse contexts.

Despite these advances in the molecular engineering of *B. bacteriovorus*, several issues remain. First, we have only examined a subset of promoters; extending this approach to include a wider range of promoter libraries, possibly guided by transcriptomic or proteomic data [32,45,70–73] may reveal additional layers of regulatory control and enable more systematic tuning of gene expression in *B. bacteriovorus*. Second, the RBS sequences used here can be further improved through rational design and large-scale combinatorial libraries, allowing deeper insights into the *B. bacteriovorus* translational machinery and accelerating the discovery of highly efficient translation initiation signals. Third, while our NanoLuc assay demonstrated Sec-dependent secretion, other *B. bacteriovorus* secretion pathways (e.g., type I or twin-arginine translocation [30]) need to be investigated in the future, specifically concerning the suitability for secretion of specific heterologous proteins. Finally, understanding how these engineered constructs perform under different environmental conditions - such as nutrient limitation or oxygen stress - will be crucial for exploiting *B. bacteriovorus* in real-world contexts, including soil, water, and clinical settings.

In summary, this work significantly expands the genetic engineering repertoire available for *B. bacteriovorus*. By validating the functionality of synthetic and native promoters, refining RBS sequences, and developing a sensitive secretion assay, we provide tools that will drive both fundamental investigations of predation biology and the advancement of predator-based biotechnologies. As interest in exploiting predatory bacteria for antimicrobial and environmental applications grows, our results highlight the importance of combining robust genetic tools, single-cell approaches, and quantitative secretion assays to fully harness the therapeutic and biocontrol potential of this remarkable organism.

## Supporting information

**S1 Fig. Overview of the *B. bacteriovorus* predatory life cycle.**
(PDF)

**S2 Fig. Plasmid map of pLH-C1 (pCAT:P$_{merRNA}$-opt.RBS-mScarletI3), derived from pCAT.000, showing the mScarletI3 reporter gene under the control of promoter P$_{merRNA}$ and an RBS sequence optimized for *B. bacteriovorus* (opt.RBS).**
(PDF)

**S3 Fig. Overview of mScarlet fluorescence distributions in cells harbouring various pCAT.000-derived vectors in *B. bacteriovorus* AP (a) and *E. coli* S17-1 (b).**
(PDF)

**S4 Fig. Native and synthetic promoters exhibit distinct mScarletI3 expression levels and variability in *E. coli* S17-1.**
(PDF)

**S5 Fig. Extended temporal gene expression of selected synthetic and native promoters in *B. bacteriovorus* populations during predation over 72 hours.**
(PDF)

**S6 Fig. Removal of Ribosomal binding site (RBS) from *B. bacteriovorus* native promoters and synthetic promoters leads to low expression level of mScarletI3 in *B. bacteriovorus* AP populations.**
(PDF)

**S7 Fig. Effect of promoter and ribosome binding site (RBS) on gene expression levels in *B. bacteriovorus* AP (a) and *E. coli* S17-1 (b).**
(PDF)

**S1 Table. Overview of promoter regions used in this study.**
(PDF)

**S2 Table. Overview of the Amino acid and DNA sequences used for Sec-dependent signal peptides fused to NanoLuc reporter gene used in this study.**
(PDF)

**S3 Table. *B. bacteriovorus* strains used in this study.**
(XLSX)

**S4 Table. *E. coli* strains used in this study.**
(XLSX)

**S5 Table. Plasmids generated and used in this study.**
(XLSX)

**S6 Table. Primers used in this study.**
(XLSX)

## Acknowledgments

The authors would like to thank Prof. Leo Eberl (University of Zurich) for hosting the main authors in his laboratory.

## Author contributions

**Conceptualization:** Ljiljana Mihajlovic, Andreas Diepold, Simona G. Huwiler.

**Data curation:** Ljiljana Mihajlovic, Simona G. Huwiler.

**Formal analysis:** Ljiljana Mihajlovic, Lara M. Hofacker.

**Funding acquisition:** Ljiljana Mihajlovic, Andreas Diepold, Simona G. Huwiler.

**Investigation:** Ljiljana Mihajlovic, Lara M. Hofacker, Florian Lindner, Simona G. Huwiler.

**Methodology:** Ljiljana Mihajlovic, Florian Lindner, Priyanikha Jayakumar, Andreas Diepold, Simona G. Huwiler.

**Project administration:** Ljiljana Mihajlovic, Simona G. Huwiler.

**Resources:** Ljiljana Mihajlovic, Florian Lindner, Priyanikha Jayakumar, Andreas Diepold, Simona G. Huwiler.

**Supervision:** Ljiljana Mihajlovic, Simona G. Huwiler.

**Validation:** Ljiljana Mihajlovic, Simona G. Huwiler.

**Visualization:** Ljiljana Mihajlovic, Simona G. Huwiler.

**Writing – original draft:** Ljiljana Mihajlovic, Simona G. Huwiler.

**Writing – review & editing:** Ljiljana Mihajlovic, Lara M. Hofacker, Florian Lindner, Priyanikha Jayakumar, Andreas Diepold, Simona G. Huwiler.

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
