## [Decision Letter · Decision Letter 0]

27 Jul 2025

PGENETICS-D-25-00685

Modulating gene expression and protein secretion in the bacterial predator Bdellovibrio bacteriovorus

PLOS Genetics

Dear Dr. Huwiler,

Thank you for submitting your manuscript to PLOS Genetics. After careful consideration, we feel that it has merit but does not fully meet PLOS Genetics's publication criteria as it currently stands. Therefore, we invite you to submit a revised version of the manuscript that addresses the points raised during the review process.

Please submit your revised manuscript within 60 days Sep 25 2025 11:59PM. If you will need more time than this to complete your revisions, please reply to this message or contact the journal office at plosgenetics@plos.org. Please include the following items when submitting your revised manuscript:

We look forward to receiving your revised manuscript.

Kind regards,

Kai Papenfort

Academic Editor

PLOS Genetics

Danielle Garsin

Section Editor

PLOS Genetics

Aimée Dudley

Editor-in-Chief

PLOS Genetics

Anne Goriely

Editor-in-Chief

PLOS Genetics

**Additional Editor Comments:**

Dear Dr Huwiler.

Thank you for submitting your work to PLOS Genetics. The manuscript has now been reviewed by three experts in the field and their comments are provided below. As you can see from their comments below, all referees find your work potentially interesting, however, they have also raised several criticisms. One main criticism concerns the novelty of the work (see comments of reviewer #2), which should be carefully addressed in the revised version of your manuscript and the rebuttal letter.

Best wishes

Kai Papenfort

(Academic Editor)

**Journal Requirements:**

- ® on pages: 19, and 20

- TM on page: 20.

3) Your manuscript is missing the following sections: Description of the Method, Verification and Comparison, and Applications. Please ensure that your article adheres to the standard Methods article layout and order of Abstract, Author Summary, Introduction, Description of the Method, Verification and Comparison, Applications, Discussion, Acknowledgements, References, and Supplementary Information. For details on what each section should contain, see our Methods article guidelines:

https://journals.plos.org/plosgenetics/s/submission-guidelines#loc-manuscript-organization.

5) We notice that your supplementary Figures, and Tables are included in the manuscript file. Please remove them and upload them with the file type 'Supporting Information'. Please ensure that each Supporting Information file has a legend listed in the manuscript after the references list.

Potential Copyright Issues:

i) Figures 5A, and S1. Please confirm whether you drew the images / clip-art within the figure panels by hand. If you did not draw the images, please provide (a) a link to the source of the images or icons and their license / terms of use; or (b) written permission from the copyright holder to publish the images or icons under our CC BY 4.0 license. Alternatively, you may replace the images with open source alternatives. See these open source resources you may use to replace images / clip-art:

7) When completing the data availability statement of the submission form, you indicated that you will make your data available on acceptance. We strongly recommend all authors decide on a data sharing plan before acceptance, as the process can be lengthy and hold up publication timelines. Please note that, though access restrictions are acceptable now, your entire data will need to be made freely accessible if your manuscript is accepted for publication. This policy applies to all data except where public deposition would breach compliance with the protocol approved by your research ethics board. If you are unable to adhere to our open data policy, please kindly revise your statement to explain your reasoning and we will seek the editor's input on an exemption. Please be assured that, once you have provided your new statement, the assessment of your exemption will not hold up the peer review process.

8) Please amend your detailed Financial Disclosure statement. This is published with the article. It must therefore be completed in full sentences and contain the exact wording you wish to be published.

9) Please ensure that the funders and grant numbers match between the Financial Disclosure field and the Funding Information tab in your submission form. Note that the funders must be provided in the same order in both places as well.

**Reviewers' comments:**

Reviewer's Responses to Questions

Reviewer #1: One focus of microbiology is on predatory bacteria in recent years. A limited factor is a scarcity of toolkits for genetic operation. In this manuscript showed that authors developed a molecular toolbox for gene expression and protein secretion. To my knowledge, it is the second system for gene expression and the first system for protein secretion to Bdellovibrionota’s bacteria. Authors showed a rational design, fine implementation for experiments, good english writing, and clear, understandable figures and tables. I appreciate this manuscript.

Here I give one suggestion for the Title. This article is to focus on the development of a genetic toolkit for predatory bacteria, which is used for gene expression and protein secretion. Thus, my suggestion is to modify the title “Modulating gene expression and protein secretion in the bacterial predator Bdellovibrio bacteriovorus” as “A molecular toolbox development to modulate gene expression and protein secretion in the bacterial predator Bdellovibrio bacteriovorus”.

Two typewriting errors:

In Line 87, “…as in E. coli, are limited [12,28] To date,” modified to “…as in E. coli, are limited [12,28]. To date,”

In Line 89, “The scarcity of versatile gene expression systems than can be fine-tuned…”, modified to “The scarcity of versatile gene expression systems that can be fine-tuned…”

Reviewer #2: Major general comments:

The experimental work presented in PGENETICS-D-25-00685 is well done, the manuscript is clear and well written, and the topic is interesting. However, I have serious doubts about the novelty and impact of the work when considering the article published in 2024 by Salgado et al., entitled “Controlling the expression of heterologous genes in Bdellovibrio bacteriovorus using synthetic biology strategies” Microbial Biotechnology, 17, e14517.https://doi.org/10.1111/1751-7915.14517. This article was cited as reference # 28 in PGENETICS-D-25-00685, exclusively in the introduction section, as one of the few previous works describing a limited number of independent replicative plasmids and chemically inducible functional promoters. However, many of the aspects described in PGENETICS-D-25-00685, and claimed to be original work, had already been developed by Salgado et al. and this work were not properly cited, significantly diminishing the novelty and importance of Mihajlovic et al.'s work.

The details of these aspects are provided below as “Specific detailed comments”.

Some aspects of PGENETICS-D-25-00685 can be considered original, such as the comparison of native versus synthetic promoters and the analysis of secretion systems in the predator. These aspects could be further developed.

Specific detailed comments that limit the novelty and impact of the work or require proper citation:

1) Development of a set of molecular tools for B. bacteriovorus through systematic fine-tuning of gene expression based on synthetic biology approaches. Although the technology applied in the two studies differs (Gibson assembly in PGENETICS-D-25-00685 versus the hierarchical assembly cloning technique Golden Standard (GS) based on the Golden Gate (GG) modular cloning used in Salgado et al), the result is similar and this is not mentioned or cited.

2) Concerning the destination vectors, both works used plasmids carrying the ori RSF1010. This is not mentioned. In addition, Salgado et al developed destination vectors that adapt SEVA plasmids with a transposon Tn7-mediated chromosomal insertion system to monitor gene expression.

3) Both studies developed a system based on fluorescent proteins to monitor gene expression by flow cytometry at the single-cell level. In this sense, Salgado et al. observed that predator cells carrying RSF1010 replicative vectors exhibited heterogeneity in terms of their fluorescent phenotype: some cells accumulated high levels of fluorescent proteins, while others accumulated lower or none levels. For this reason, these authors applied a monocopy chromosomal insertion system to monitor gene expression. Similar behaviour was observed in the results presented in Figures 1, 2, 4 and S3-S4, but this is not mentioned in the text.

4) Both works monitor gene expression in the predator using synthetic promoters from the Anderson Collection. In one case, they even use the same promoter (PJ23102 in PGENETICS-D-25-00685, named PJ102 in Salgado et al.). I notice a serious lack of novelty here.

5) Regarding the discussion about the feasibility of expressing synthetic promoters in the predator as a consequence of the existence of the appropriated σ factors in B. bacteriovorus, it was addressed here and also before in Salgado et al, but not mentioned in PGENETICS-D-25-00685.

Other comments:

6) The comparison between synthetic and native promoters is interesting and should be developed further, but highlighting previous works and describing the novel aspects of the study.

7) The section describing the evaluation of different ribosomal binding sites to fine-tune gene expression is interesting and novel, but a broader range of RBS must be considered to improve the significance of the results.

8) I found the section describing the analysis of different native Sec-dependent signal sequences very original. Perhaps this section could be further developed to enhance the novelty and impact of the work.

Reviewer #3: The authors describe their contribution to the B. bacteriovorus genetic toolbox, including the systematic assessment of RBS's, synthetic and natural promoters and the development of a nanoluc-based secretion assay. Previously, several promoters were tested and introduced for B. bacteriovorus, but the toolbox is still very limited. This study is an addition to the field, especially as the authors used single-cell methods (FACS) to test their promoters behavior in attack phase.

The study employs the nanoluc system in B. bacteriovorus for the first time to assess secretion and the efficacy of secretion tags.

The study is carried out well and a welcome addition to the still limited B. bacteriovorus toolbox. I have only few comments:

1) In the last years, several studies introduced synthetic and non-native promoters in B. bacteriovorus. It would have been a nice addition to the study to include these already used promoters (for instance Pbiofab (Kaljevic et al, 2021), PexD (Salgado et al, 2024)) as a benchmark/comparison.

2) It is notoriously difficult to do single-cell analysis on B. bacteriovorus during the growth phase, so it is understandable that the authors did not add heterogeneity studies to their promoter assessment in growth phase. However, would it not be possible to at least make a qualitative (and possibly also quantitative) assessment of the single cell behavior and promoter strength in GP using fluorescence microscopy?

**Have all data underlying the figures and results presented in the manuscript been provided?**

Reviewer #1: Yes

Reviewer #2: Yes

Reviewer #3: None

PLOS authors have the option to publish the peer review history of their article (what does this mean? ). If published, this will include your full peer review and any attached files.

**Do you want your identity to be public for this peer review?** For information about this choice, including consent withdrawal, please see our Privacy Policy .

Reviewer #1: No

Reviewer #2: No

Reviewer #3: No

**Figure resubmission:**
---

## [Decision Letter · Decision Letter 1]

21 Oct 2025

Dear Dr Huwiler,

We are pleased to inform you that your manuscript entitled "A molecular toolbox to modulate gene expression and protein secretion in the bacterial predator *Bdellovibrio bacteriovorus* " has been editorially accepted for publication in PLOS Genetics. Congratulations!

Yours sincerely,

Kai Papenfort

Academic Editor

PLOS Genetics

Danielle Garsin

Section Editor

PLOS Genetics

Aimée Dudley

Editor-in-Chief

PLOS Genetics

Anne Goriely

Editor-in-Chief

PLOS Genetics

BlueSky: @plos.bsky.social

Comments from the reviewers (if applicable):

Dear Dr Huwiler.

Thanks again for submitting your manuscript to PLOS Genetics. I am happy to inform you that all referees were very positive about the revised work and I agree with their opinions. Congrats on a very nice manuscript.

Reviewer #1 has two very minor comments (see below). Please address these when you receive the proofs of your manuscript.

Best wishes

Kai Papenfort

Reviewer's Responses to Questions

**Comments to the Authors:**

Reviewer #1: P523-525, “In contrast, pairing the native B. bacteriovorus promoter Pbd0149 with the synthetic RBS led to a decrease in mScarlet fluorescence compared to when the same promoter was combined with the species-optimized RBS” could be changed to “In contrast, the native B. bacteriovorus promoter Pbd0149 with the synthetic RBS led to a decrease in mScarlet fluorescence compared to the same promoter that was combined with the species-optimized RBS”.

P668-669: “…, which may explain why B. bacteriovorus RpoD can recognize and drive expression from synthetic σ 70 dependent E. coli promoters can be recognized by [27]” could have some grammatic error.

Reviewer #2: I have no further comments

Reviewer #3: Thanks to the authors for their constructive responses to the reviewer's comments. I have no further suggestions.

**Have all data underlying the figures and results presented in the manuscript been provided?**

Reviewer #1: Yes

Reviewer #2: Yes

Reviewer #3: Yes

PLOS authors have the option to publish the peer review history of their article (what does this mean? ). If published, this will include your full peer review and any attached files.

**Do you want your identity to be public for this peer review?** For information about this choice, including consent withdrawal, please see our Privacy Policy .

Reviewer #1: **Yes: ** Xianwu Guo

Reviewer #2: No

Reviewer #3: No

**Data Deposition**

http://datadryad.org/submit?journalID=pgenetics&manu=PGENETICS-D-25-00685R1

**Press Queries**

---

## [Editor Report · Acceptance letter]

PGENETICS-D-25-00685R1

A molecular toolbox to modulate gene expression and protein secretion in the bacterial predator *Bdellovibrio bacteriovorus*

Dear Dr Huwiler,

We are pleased to inform you that your manuscript entitled "A molecular toolbox to modulate gene expression and protein secretion in the bacterial predator *Bdellovibrio bacteriovorus* " has been formally accepted for publication in PLOS Genetics! Your manuscript is now with our production department and you will be notified of the publication date in due course.

With kind regards,

Anita Estes

PLOS Genetics

On behalf of:
